# Two-Channel Indirect-Gap Photoluminescence and Competition between the Conduction Band Valleys in Few-Layer MoS_2_

**DOI:** 10.3390/nano14010096

**Published:** 2023-12-30

**Authors:** Ayaz H. Bayramov, Elnur A. Bagiyev, Elvin H. Alizade, Javid N. Jalilli, Nazim T. Mamedov, Zakir A. Jahangirli, Saida G. Asadullayeva, Yegana N. Aliyeva, Massimo Cuscunà, Daniela Lorenzo, Marco Esposito, Gianluca Balestra, Daniela Simeone, David Maria Tobaldi, Daniel Abou-Ras, Susan Schorr

**Affiliations:** 1Institute of Physics, Ministry of Science and Education, Baku Az1143, Azerbaijan; bayramov@physics.science.az (A.H.B.); elvin.alizada@physics.science.az (E.H.A.); s.asadullayeva@physics.science.az (S.G.A.); 2Institute of Physical Problems, Baku State University, Ministry of Science and Education, Baku Az1148, Azerbaijan; yeganaaliyeva.n@bsu.edu.az; 3National Research Council, Institute of Nanotechnology (NANOTEC), University c/o Campus Ecotekne, Via per Monteroni, 73100 Lecce, Italy; daniela.lorenzo@nanotec.cnr.it (D.L.); marco.esposito@nanotec.cnr.it (M.E.); gianluca.balestra@nanotec.cnr.it (G.B.); daniela.simeone@nanotec.cnr.it (D.S.); david.tobaldi@nanotec.cnr.it (D.M.T.); 4Department of Mathematics and Physics ‘‘Ennio De Giorgi”, University of Salento, c/o Campus Ecotekne, Via per Monteroni, 73100 Lecce, Italy; 5Helmholtz-Zentrum Berlin for Materials and Energy (HZB), Department of Structure and Dynamics of Energy Materials, 14109 Berlin, Germany; daniel.abou-ras@helmholtz-berlin.de (D.A.-R.); susan.schorr@helmholtz-berlin.de (S.S.); 6Institute of Geological Sciences, Free University of Berlin, 14195 Berlin, Germany

**Keywords:** MoS_2_, sulfurization, MoO_3_, dielectric function, photoluminescence, confocal Raman spectroscopy

## Abstract

MoS_2_ is a two-dimensional layered transition metal dichalcogenide with unique electronic and optical properties. The fabrication of ultrathin MoS_2_ is vitally important, since interlayer interactions in its ultrathin varieties will become thickness-dependent, providing thickness-governed tunability and diverse applications of those properties. Unlike with a number of studies that have reported detailed information on direct bandgap emission from MoS_2_ monolayers, reliable experimental evidence for thickness-induced evolution or transformation of the indirect bandgap remains scarce. Here, the sulfurization of MoO_3_ thin films with nominal thicknesses of 30 nm, 5 nm and 3 nm was performed. All sulfurized samples were examined at room temperature with spectroscopic ellipsometry and photoluminescence spectroscopy to obtain information about their dielectric function and edge emission spectra. This investigation unveiled an indirect-to-indirect crossover between the transitions, associated with two different Λ and K valleys of the MoS_2_ conduction band, by thinning its thickness down to a few layers.

## 1. Introduction

Group-VI layered two-dimensional (2D) transition metal dichalcogenides (TMDs) (e.g., MoS_2_, WS_2_, MoSe_2_ and WSe_2_) exhibit very interesting semiconducting properties and are attracting a lot of attention that has been increasingly growing after the discovery of metallic two-dimensional graphene. As graphene, TMD monolayers have hexagonal symmetry and show a direct bandgap in the range of 1–2 eV [1]. Bi- or multilayers of TMDs are indirect bandgap semiconductors [1]. The remarkable versatility of monolayer and multilayer TMDs as a viable alternative to graphene [2] arises from their unique crystal structure. Their bandgaps and chemical properties [3] can be effectively tuned using different approaches (varying the number of layers, intercalation, alloying, mechanical stress, doping, etc.) [4,5].

As far as MoS_2_ is concerned, numerous studies have revealed its intricate electronic spectrum and optical properties, surpassing the limitations of the conventional one-electron band structure [6]. The substantial influence of multiexcitonic effects has highlighted the importance of thorough experimental studies on electronic excitations [7,8,9,10,11], placing them at the forefront of the current scientific research concerning MoS_2_. According to a series of theoretical works [1,2,3,4,5,12], the reported tunability of the direct bandgap due to indirect-to-direct crossover [13,14] in atomically thin MoS_2_ is directly related to the variable orbital composition of the involved electronic states. For the same reason, the indirect bandgap of few-layer MoS_2_ will also vary with the thinning of its thickness, and even an indirect-to-indirect crossover between the transitions associated with two different valleys of the conduction band may occur. While few-layer MoS_2_ exhibits two valleys along the Γ–K line with similar energy, as highlighted by Zhao et al. [15], the particular valley responsible for forming the conduction band minimum remains poorly understood. Fundamental questions persist regarding the circumstances under which the indirect Λ−Γ transition will shift to indirect K−Γ band alignment and whether this shift will occur at all, because the valley responsible for forming the conduction band minimum is yet to be specified.

Until now, MoS_2_ monolayer and few-layer films were obtained using the exfoliation technique [10,13,16,17], chemical vapor deposition (CVD) [11,18,19,20,21,22,23], the sulfurization of CVD-deposited MoCl_5_ [24] and MoO_3_ [25] and the sulfurization of Mo [26] and MoO_3_ [27] thin films obtained using vapor phase growth.

In this work, we report the preparation of MoS_2_ ultrathin films by sulfurization of few-layer MoO_3_ films that were preliminary obtained on a SiO_2_/Si substrate by plasma-enhanced atomic layer deposition (PE-ALD). All sulfurized samples were examined at room temperature with spectroscopic ellipsometry and photoluminescence spectroscopy to obtain information about their dielectric functions (DF) and emission spectra. The latter unveiled the competition of two indirect transitions from the Λ and K valleys of the conduction band by thinning the MoS_2_ thickness down to a few layers.

## 2. Experimental Section

### 2.1. MoO_3_ Thin Films

The MoO_3_ films were deposited in a PE-ALD system (SI ALD LL, SENTECH Instruments, Berlin, Germany), as described in detail in [28,29]. The films were grown on thermal oxidized (SiO_2_ with a nominal thickness of 105 nm) p-type Si (100) wafers (thickness ~ 650–700 µm) with a resistance of <0.005 Ohm × cm (high-doped substrates).

X-ray diffraction (XRD) patterns were recorded on a Malvern PANalytical X’Pert Pro MRD diffractometer(Malvern Panalytical Ltd., Bristol, UK) equipped with a fast PIXcel detector, using CuK*_α_* radiation generated at 40 kV and 40 mA. Grazing incidence XRD (GIXRD) patterns were recorded at an incident angle of 0.5°, with a step size of 0.01 °2θ and a counting time of 0.5 s, over a 5–55 °2θ interval. Typical GIXRD patterns and dielectric functions (DFs) retrieved using spectroscopic ellipsometry (SE), performed with the aid of a rotating-compensator M 2000DI (J.A. Woollam, Lincoln, NE, USA) ellipsometer at different incident angles over the photon energy range of 0.7–6.5 eV, are given in Appendix A and Appendix A, respectively. All the obtained MoO_3_ films, subjected to further sulfurization, were crystalline, and their DFs were indicative of the absence of the oxygen deficiency, as discussed in the Appendix A. Their thicknesses, found in an X-ray reflectometry (XRR) examination were 3, 5 and 30 nm.

### 2.2. MoS_2_ Thin Films

The sulfurization of the MoO_3_ films was carried out in a 2-inch single-zone tube furnace (OTF-1200X-S, MTI Corporation, Richmond, CA, USA). For this purpose, the MoO_3_ films were placed at the center of the furnace. The furnace and samples were purged multiple times under Ar flow (250 sccm 99.999% Ar). The temperature of the furnace was ramped up from room temperature to 700 °C over 15–20 min and maintained at that temperature for 60 min under atmospheric pressure. During heating, H_2_S (10 sccm 99.5% H_2_S) was introduced at 300 °C; it was removed at the end of the 60 min thermal treatment. Afterward, the furnace was purged with Ar (250 sccm) during the cool-down process. After cooling to ~400 °C, the furnace was opened for rapid cooling at ~100 °C over ~10 min.

The topographies of the prepared 30, 5 and 3 nm-thick MoS_2_ films (plan view) were examined using a Zeiss UltraPlus scanning electron microscope (Carl Zeiss, Oberkochen, Germany). The corresponding scanning electron microscopy (SEM) images are given in Appendix A.

Examinations using X-ray reflectivity and SE revealed that sulfurization induces changes in film thickness. Specifically, the thickness of 30 nm-thick film was reduced by nearly 30% after sulfurization. On the other hand, the films with nominal thicknesses of 3 and 5 nm each experienced less than a 15% change in thickness after sulfurization. To simplify the references in this text, each of the MoS_2_ films studied will be identified based on its nominal thickness, which corresponds to the thickness of the source MoO_3_ film before sulfurization. The variation in thickness among the resulting MoS_2_ films is distinctly evident in their Raman spectra (Appendix A). These were recorded using back-scattering geometry on a Nanofinder-30 confocal Raman system (Tokyo Instrument Inc., Tokyo, Japan) equipped with a Juno 3050 GS-11 (Kyocera Soc Corporation, Yokohama, Japan) Nd:yttrium–aluminum–garnet laser (second harmonic, 532 nm). The maximum output power of the excitation source was 10 mW. The cross-sectional beam diameter was 4 μm. Diffraction grating with 1800 grooves per mm provided a spectral resolution of 0.5 cm^−1^. The spectral signal was detected using a photon-counting charge-coupled device (CCD) camera “Andor” (Andor Technology, Belfast, Ireland) cooled down to −100 °C.

As shown in Appendix A, the observed decreases in intensity and disappearance of the 521 cm^−1^ Raman line of the Si substrate with increasing thickness of the MoS_2_ film (Appendix A) corroborates the value of the absorption coefficient (Appendix A) extracted from the DFs of the 3, 5 and 30 nm MoS_2_ films.

Despite the large number of theoretical works on ultrathin MoS_2_, only a few address its DFs. So far, as the thicknesses of the obtained 3 and 5 nm MoS_2_ samples exceeded one layer (L) and approximately corresponded to 4 L and 8 L MoS_2_ (Appendix A), respectively, they were of primary importance in the context of the present work. Nevertheless, along with the latter two, bulk and 2 L MoS_2_ were also included in the band structure calculations to obtain a more complete picture and unravel the main trends that band structure and DF show upon thinning of the MoS_2_ thickness.

We used the full-potential linearized augmented plane wave (FP LAPW) method implemented in the scheme reported in [30]. The exchange-correlation interactions were described as within the generalized gradient approximation (GGA), using the strategy reported in [31]. The convergence parameter R_mt_K_max_, where R_mt_ is the smallest atomic sphere radius and K_max_ is the largest K-vector of the plane wave expansion of the wave function, was set to 7.0. Within the atomic spheres, the partial waves were expanded up to l_max_ = 10, where l_max_ is the highest value of the orbital angular-momentum quantum number used for partial waves inside atomic spheres. Integrations over the first Brillouin zone (BZ) were performed using the tetrahedron method, with 60 points in the irreducible part of the BZ. The R_mt_ values for Mo and S were set to 2.34 and 2.08 a. u. (atomic unit), respectively. The value of −6.0 Ry of cut-off energy was used for the separation of the core and valence states. The imaginary part of the DF was calculated using the joint density of the states for optical transitions between the valence and conduction bands, using the Monkhorst–Pack technique for integration over the Brillouin zone [32]. The real part of the dielectric function was calculated from the imaginary part using the Kramers–Kronig relation.

While SE is a powerful tool for studying direct optical transitions, photoluminescence (PL) is a well-endorsed technique for studying indirect-gap emissions in MoS_2._

In the present work, PL studies were performed on an Infrared PL/PLE/Raman spectrometer (Tokyo Instrument Inc., Tokyo, Japan) using lasers with two excitation wavelengths, 785 nm (NovaPro 785–250) and 532 nm (NovaPro PB 532–200 DPss), to embrace a possibly wider range of intrinsic electronic excitations that may decay radiatively and contribute to the PL of MoS_2_. The obtained PL spectra are given and discussed together with the other main results in the next section.

## 3. Results and Discussion

Among the significant parameters in the resulting electronic energy spectrum of MoS_2_ (Figure 1a–d) is the splitting between adjacent bands at the K-point of the BZ. As highlighted in Figure 1, for each MoS_2_ layer count, it arises from interlayer interactions and spin–orbit coupling (SOC). The splitting magnitude was reduced in the 8, 4 and 2 L MoS_2_ compared to the bulk material, primarily due to the finite number of layers in ultrathin MoS_2_. In a 1 L MoS_2_ limit, the splitting solely originates from SOC [3].

As shown in Figure 1e (shaded area), this splitting manifests itself through the irregular behavior of the ellipsometric parameter Ψ within the narrow photon energy gap spanning from 1.8 to 2 eV.

Such peculiar behavior was observed across all MoS_2_/SiO_2_/Si stacks studied in this work and for all incident angles accessed during the SE measurements. Note that the mean square error (MSE) given in Figure 1e for each studied MoS_2_/SiO_2_/Si structure is related to the model that was fitted to the ellipsometric parameters not only within the 1.4 to 2.1 eV range shown in Figure 1 but also for the entire accessible photon energy range spanning from 0.7 to 6.5 eV. The obtained MSE values fell below eight, indicating that the fit was acceptable and the retrieved DF was accurate enough.

The DF across the entire range of photon energies is displayed in Figure 2, with separate representation for the real (a) and imaginary (b) parts. The data for the MoS_2_ films with thicknesses of 30, 5, and 3 nm are indicated by the blue, green and red lines, respectively.

Based on its crystal structure and symmetry [33], MoS_2_ is a uniaxial crystal, and its optic axis is normal to the layer plane (LP). As reported in Appendix A, the calculated DF indeed shows different dispersions for light polarized parallel and perpendicular to the layer plane. Comparison between the SE-based (Figure 2) and calculated (Appendix A) data drove us to the conclusion that the obtained SE-based DF is overwhelmingly determined by dielectric response to the light polarized in the LP for all MoS_2_ films studied in this work. The descending trend in the intensity of the main features of the SE-based DFs of the 30, 5 and 3 nm MoS_2_ was reproduced using the calculated DFs obtained before and after rescaling (Appendix A). The latter mitigated the vacuum dilution effect, since the band structure calculations for MoS_2_ with a finite number of layers were conducted using supercells that included vacuum space.

The A, B and C transitions depicted in Figure 1 and reproduced by SE-based DFs in Figure 2 are commonly observed in the majority of the related studies. The exciton peaks (Figure 2) related to these transitions clearly exhibit blue shifts with the thinning of the MoS_2_ samples. The E-exciton, which is also frequently observed in optical studies [34] does not show noticeable shift. The most significant shift was observed for the C-excitons, which are not originating from transitions between parabolic bands with opposite concavities, like A- and B-excitons, but resulting from transitions between the valence and conduction band regions with similar concavities, known as the nested band regions. The comparison of the C-exciton position in 30 nm MoS_2_ with those in 5 and 3 nm MoS_2_ indicated a considerable blue shift upon thinning, exceeding 100 meV (Figure 2b). As illustrated in the inset of Figure 2b, the A- and B-excitons also exhibited shifts toward higher photon energies as the thickness of the obtained MoS_2_ was reduced to 5 or 3 nm. In comparison to the C-excitons, the B- and A-excitons experienced smaller shifts (approximately 50 and 20 meV, respectively).

Overall, as stated before, the above analysis confirms that the obtained 30 nm MoS_2_ is a good counterpart to bulk MoS_2_. Along with the already mentioned Raman spectra (Appendix A), this assertion was corroborated by the PL spectra taken for the studied MoS_2_ thin films and shown in Appendix A.

The exciton landscape in MoS_2_ is highly intricate, embracing neutral, charged and dark excitons, all of which directly or indirectly contribute to the DFs. This landscape is dynamic, with various components influenced by numerous factors, including the fabrication process [20]. Therefore, when the retrieved DFs are evaluated, they should be analyzed in conjunction with PL data, comparing the absorption coefficient derived from the DFs to the PL spectra within the same spectral range. Since our primary focus was on few-layer MoS_2_, we initially concentrated on MoS_2_/SiO_2_/Si structures featuring 3 and 5 nm-thick MoS_2_ layers.

Figure 3 displays the photon energy dependencies of the absorption coefficients (α) and PLs of the few-layer MoS_2_ films. The PLs under the excitation wavelength of 532 nm (2.33 eV) are given for various levels of excitation power. The normalized PL spectrum (shape function) for each excitation level underwent little change, with increasing excitation power in the range of 2–10 mW (Appendix A), and the positions of the emission lines remained unchanged.

A- and B-excitons, which are positioned above 1.8 eV in MoS_2_ [13,33,34,35,36], clearly manifested themselves in the spectral features of the α and PL values of the studied films. Comparison with the reported energy positions of A_exc._ (1L MoS_2_) and B_exc._ (1L MoS_2_) for a single layer showed that the energy gap between the A- and B-excitons in our case did not exceed approximately 150 meV. This value is noticeably smaller than 200 meV that is the value of the splitting between A- and B-excitons in bulk MoS_2_ [3]. This observation is directly related to a few layers’ thickness of the prepared films. In the case of the 5 nm MoS_2_, the energy position of the emission line ascribed to the B-excitons was red-shifted by nearly 20 meV from its energy position in the MoS_2_ monolayer and from its absorption peak as reported in Figure 2b. For the 3 nm MoS_2_, the shift was definitely less pronounced (see Figure 2a). The energy difference between absorption and emission, known as the Stokes shift, is inherent to all materials and varies widely depending on the material type (semiconductor, molecular crystal) and internal electrical fields. For typical semiconductors like GaAs, this shift is only 4 meV [39]. There are no reports on the intrinsic Stokes shift value in MoS_2_. However, as the shift is decreased in 3 nm MoS_2_ compared to 5 nm MoS_2_, the Stokes shift in the latter is non-intrinsic and is likely caused by some strain or inhomogeneity in 5 nm MoS_2_.

Strong enhancement in PL intensity related to the direct exciton radiative transitions in MoS_2_ when MoS_2_ thickness goes down to a few nanometers is a well-known and solid fact. According to the PL spectra in Figure 3, some enhancement also takes place at the nanoscale when MoS_2_ thickness is changed from 5 to 3 nm. It can be clearly seen that under the same excitation levels, the PL intensity in 3 nm MoS_2_ was noticeably stronger than in the 5 nm MoS_2_.

While α is structureless and, for both the 3 and 5 nm MoS_2_, showed only absorption tails descending toward lower photon energies, the PL spectrum showed an indirect-gap emission band centered around 1.25 eV (Figure 3).

It has been established in many experimental and theoretical works [2,5,14,15,18,38,40,41] that MoS_2_ remains an indirect semiconductor even when reduced to bilayers. Observation of the indirect-gap emission around 1.25 eV on 3 nm-thick MoS_2_ (Figure 3) has revealed that MoS_2_ retains its indirect characteristics even when its thickness is reduced to just four layers. This agrees well with the already-quoted works.

However, as noted by Zhao et al. [15], few-layer MoS_2_ exhibits two valleys along the Γ–K line with similar energy, and there is limited understanding of which valley forms the conduction band minimum. It is important to highlight that the indirect-gap value of 1.20 eV [38] characterizes bulk MoS_2_ and corresponds to the I_1_ transition in the band structure shown in Figure 1. This implies that, like in previous calculations, including the cited work [15], the band structure in Figure 1 captures the main trends in the evolution of band structure with the thinning of MoS_2_. It indicates a higher energy of K–Γ transitions (I_2_) compared to Λ–Γ transitions (I_1_) and even predicts a crossover between the Λ and K conduction band valleys. However, the precise temperature at which this crossover occurs in few-layer MoS_2_ is beyond the predictive capabilities of the performed calculations. Nevertheless, discovering experimental conditions that would enable the simultaneous generation of both I_1_ and I_2_ radiative transitions at room temperature would greatly assist in addressing this issue.

The PL spectra displayed in Figure 3 were excited with a 532 nm (2.33 eV) laser and, along with direct exciton lines, showed only indirect I_1_ (Λ–Γ) emission. The photon energy of 2.33 eV considerably exceeds the direct energy gap between the conduction and valence bands at the K point, and this may have prevented excited carriers from emitting I_2_ (K-Γ).

Note that some relaxation paths for excited electrons can be blocked or very slow before emission, as has been observed in a case of high energy C-excitons in MoS_2_ [42]. However, considering the blocking of carriers excited at 2.33 eV from relaxation down to the K–point to prevent K–Γ indirect emission seems implausible. Such a blocking mechanism would contradict the observation of the intense direct gap exciton emission (Figure 3) associated with the same K point in the band structure (Figure 1). It is more reasonable to assume that the direct emission channel for the K-point is considerably more effective than the indirect emission channel and the K–Γ indirect emission does not manifest itself under 2.33 eV excitation. This assumption has received strong experimental evidence in the PL spectra taken under excitation with a wavelength of 785 nm or photon energy of 1.57 eV, which was below the direct bandgap energy to exclude the excitation of direct emission.

Indeed, as shown in Figure 4a, b for the 3 and 5 nm MoS_2_, respectively, both the I_1_ (Λ–Γ) and I_2_ (K–Γ) indirect-gap emissions, with experimental energies corresponding to 1.25 and 1.4 eV, respectively, were observed in the PL spectra under 1.57 eV excitation.

It is important to stress that although both the Λ and K valleys were involved in the indirect emission of the obtained few-layer MoS_2_, the PL spectra of the 3 (Figure 4a) and 5 nm (Figure 4b) MoS_2_ differed in some details.

Along with the emission line related to the K–Γ transition, I_2_ and besides the emission line associated with the Λ-Γ transition, I_1_, the room-temperature PL spectra of the 3 nm MoS_2_ also showed a small satellite (I_1_+50 meV) positioned 50 meV higher than I_1_ (Figure 4a). On the other hand, the K–Γ emission I_2_ in the 5 nm MoS_2_ was split into two components, denoted as I_2_ and I_2_–42 meV, while the I_1_+50 meV satellite was no longer seen in the spectra (Figure 4b).

The disparity observed in the PL between the 3 and 5 nm-thick MoS_2_ highlights the complicated and thickness-dependent character of the radiative processes in ultrathin MoS_2_. Further investigation is required to understand their origin. However, when the intensity ratio between the I_1_ and I_2_ emissions was compared under similar excitation powers (11.46 mW for the 3 nm MoS_2_ and 11.5 mW for the 5 nm MoS_2_), the results strongly suggested that at room temperature, 3 nm (4 L) MoS_2_ is more likely to be closer to the K-Λ crossover than 5 nm (8 L) MoS_2_. Preliminary studies on temperature-dependent PL further support this assumption.

Until now, the concurrent generation of PL related to both the I_1_ and I_2_ indirect transitions had not been detected in MoS_2_ thin films. In a work by Luo et al. [43], the simultaneous generation of similar indirect-gap emissions was reported in multilayer MoS_2_ bubbles prepared from exfoliated MoS_2_ thin films. However, the exfoliated thin films themselves did not exhibit any PL [43]. This effect was achieved through the introduction of surface strain in multilayer bubbles and has little in common with our case. Lastly, it is essential to note that the simultaneous observation of indirect-gap emissions I_1_ (Λ-Γ transitions) at 1.25 eV and I_2_ (K-Γ transitions) at 1.4 eV occurred exclusively under excitation with a photon energy of 1.57 eV, which is below the energy gap for direct transitions in MoS_2_. For excitation with a photon energy of 2.33 eV or higher, beyond the energy gap for direct transitions in MoS_2_, the K-Γ indirect emission channel would become ineffective.

## 4. Conclusions

The sulfurization process applied to ultrathin MoO_3_ initially deposited on SiO_2_/Si substrates using plasma-enhanced atomic layer deposition has transformed MoO_3_ into ultrathin MoS_2_. The achieved material, in the form of a few layers, was deeply investigated using spectroscopic ellipsometry and photoluminescence with different excitation energies, supported by first-principles DFT-based calculations. This study has given rise to the first observation of the simultaneous generation of the indirect-gap PL caused by the indirect radiative transitions involving two distinct valleys (Λ and K) within the conduction band. This discovery provides a fresh insight into the electronic band structure of ultrathin MoS_2_ and a new platform for experimental studies into its bandgap.

## Figures and Tables

**Figure 1 nanomaterials-14-00096-f001:**
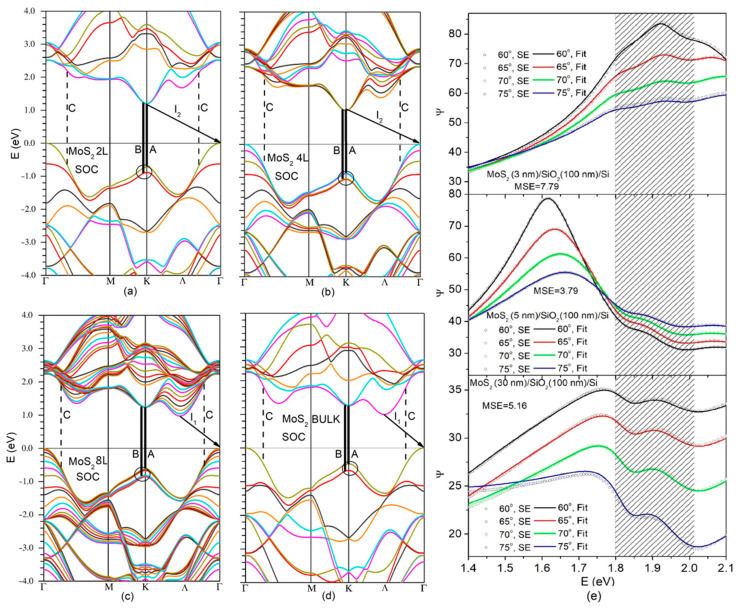
Electronic band structures of 2 L (**a**), 4 L (**b**), 8 L (**c**) and bulk (**d**) MoS_2_. Valence band splitting is encircled; possible indirect radiative transitions are indicated by arrows; and bold and dashed vertical lines show direct transitions A, B and C, respectively. (**e**) Ellipsometric parameter Ψ as a function of photon energy for the obtained MoS_2_/SiO_2_/Si structures with 3 (top plot), 5 (middle plot) and 30 nm (bottom plot) MoS_2_. On the top, colored open circles and solid lines represent experimental data and the fits to these data, respectively; each color corresponds to one of the four accessed angles of incidence.

**Figure 2 nanomaterials-14-00096-f002:**
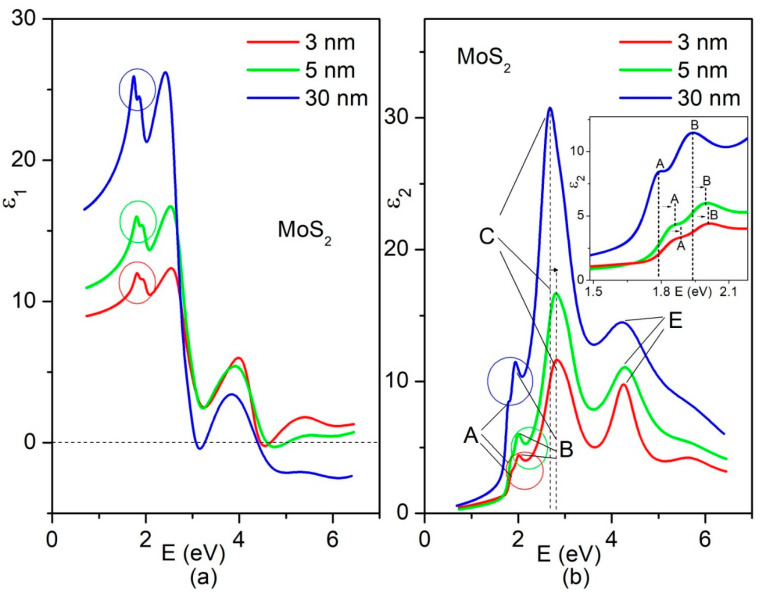
Real (**a**) and imaginary (**b**) parts of the experimental DFs retrieved for 30 (blue lines), 5 (green lines) and 5 nm (red lines)-thick MoS_2_ in a wide photon energy range. The splitting-related features of the spectra are encircled. Notations A, B, C and E, commonly used for MoS_2_, are related to the particular peaking structures in the spectrum of each of the considered MoS_2_ thin films. The vertical dashed lines and small horizontal arrows are given for convenience to show how A, B, and C exciton peak positions shift with changing the thickness of MoS_2_. As illustrated in the inset in (**b**), A- and B-excitons also exhibited shifts toward higher photon energies as the thickness of the obtained MoS_2_ was reduced to 5 or 3 nm.

**Figure 3 nanomaterials-14-00096-f003:**
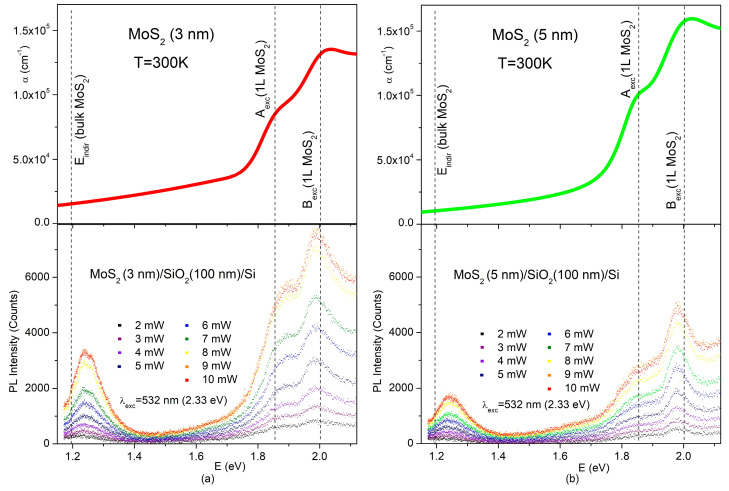
Top part: SE-based absorption coefficients of the 3 (**a**) and 5 nm (**b**) MoS_2_, respectively; bottom part: PL spectra at various excitation densities under the excitation wavelength of 532 nm (2.33 eV) for MoS_2_/SiO_2_/Si thin film structures with 3 (**a**) and 5 nm (**b**) MoS_2_ on the top. Vertical dashed lines, given for eye guidance, show the reported positions of the A- and B-excitons in a single layer of MoS_2_ [13,34,35,36,37] and the reported value of the indirect gap for bulk MoS_2_ [38].

**Figure 4 nanomaterials-14-00096-f004:**
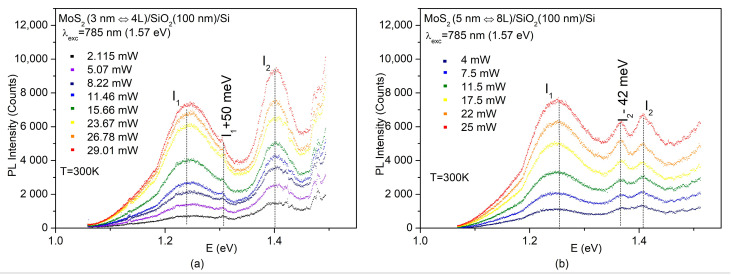
PL spectra at various excitation densities under the excitation wavelength of 785 nm (1.57 eV) for MoS_2_/SiO_2_/Si thin-film structures with 3 (**a**) and 5 nm (**b**) MoS_2_ on the top, respectively.

## Data Availability

These data are available upon request.

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
