# Peer review of "Two-Channel Indirect-Gap Photoluminescence and Competition between the Conduction Band Valleys in Few-Layer MoS2"

_nanomaterials, 2023, doi:10.3390/nano14010096_

Round 1

Reviewer 1 Report

Comments and Suggestions for Authors

This paper reports the preparation of 3 nm, 5 nm and 30 nm MoS2 thin films by sulfidation process and plasma-enhanced atomic layer deposition. All the sulfide samples were studied in depth by combining spectroscopic ellipsometry, photoluminescence and band calculation. The information of their dielectric function and edge emission spectra was obtained, which revealed the indirect to indirect crossover between the transitions associated with two different Λ and K valleys of the MoS2 conduction band. Therefore, the topic of this work is interesting, but there are still some problems in the description and analysis of the experiment in this paper, and this paper need be revised majorly. I have some questions and comments, but are not limited to:

1. There is a lack of explanation of the experimental phenomenon in the article. For example, Fig.2 only describes the observed features A, B, C, and E 155 exciton peaks, but does not explain why the test was done and what it is hoped to illustrate with this test.

2. In this paper, the band structure of 2-layer, 4-layer, 8-layer and bulk MoS2 is calculated, but the sample thickness prepared in the experiment is 3nm, 5nm, 30nm, and the layer thickness of the two is not one-to-one corresponding, So what is it trying to illustrate when you put them together? Or can the two corroborate each other?

3.When performing photoluminescence tests on samples of different powers, it is pointed out that the energy position of the B-exciton is redshifted by 20mev, so which layer of MoS2 occurs this phenomenon and what causes the redshift of energy, which needs to be explained in detail by the authors.

4.There are some non-standard scientific terms in the group of words, such as "the few monolayer films with initial thicknesses of 3 and 5 nm" at line 100, “the few monolayer ”group has some ambiguity, the paper is prepared 3nm, 5nm and 30nm layer thickness MoS2, that is, the few layers of MoS2, the use of few monolayer is easy to misunderstand as a single layer.

Comments on the Quality of English Language

The overall quality of English is OK.

Reviewer 2 Report

Comments and Suggestions for Authors

The study investigates the interlayer interaction and thickness-dependent properties of MoS2 by examining the sulfurization process of MoO3 thin films ranging from a nominal thickness of 30 nm down to a few layers. The authors argue that there is limited study for thickness-induced evolution or transformation of the indirect bandgap.

I disagree with the author's assertion, as the transition from direct to indirect bandgap due to thickness variation is well-documented in numerous prior studies. Therefore, if the novelty claimed by the author pertains solely to the electronic and optical structure changes dependent on the layering of MoS2, it has already been extensively reported in existing literature.

However, from my perspective, the novelty in this work lies in presenting experimental and theoretical findings regarding the sulfurizated MoO3 thin films, specifically yielding the thickness dependant MoS2. Viewing it in this point, this research contributes to the understanding of this process.

Considering this aspect, with appropriate revisions, this work could be suitable for publication in nanomaterials. In this following, I leave some question and comment for further improvement.

(1) In Figure S2.1a, the Raman peak position shows no change with varying MoS2 thickness, which contradicts numerous previous reports. The author should provide a detailed explanation for this discrepancy.

(2) Figure S2.1d does not appear to depict 3 nm-thick MoS2 layers; the surface looks uneven. This discrepancy should be addressed and explained.

(3) Regarding Figure 4, it is strongly recommended to include a comparison of photoluminescence (PL) spectra between reported monolayer MoS2 and bulk MoS2 with sulfurized MoO3 thin films to substantiate the authors' claim

(4) minor: check the arragement of figures and axis text, partcularly Fig. 1. Figures are not aligned. it make this figure unprofessional. 

Comments on the Quality of English Language

The study investigates the interlayer interaction and thickness-dependent properties of MoS2 by examining the sulfurization process of MoO3 thin films ranging from a nominal thickness of 30 nm down to a few layers. The authors argue that there is limited study for thickness-induced evolution or transformation of the indirect bandgap.

I disagree with the author's assertion, as the transition from direct to indirect bandgap due to thickness variation is well-documented in numerous prior studies. Therefore, if the novelty claimed by the author pertains solely to the electronic and optical structure changes dependent on the layering of MoS2, it has already been extensively reported in existing literature.

However, from my perspective, the novelty in this work lies in presenting experimental and theoretical findings regarding the sulfurizated MoO3 thin films, specifically yielding the thickness dependant MoS2. Viewing it in this point, this research contributes to the understanding of this process.

Considering this aspect, with appropriate revisions, this work could be suitable for publication in nanomaterials. In this following, I leave some question and comment for further improvement.

(1) In Figure S2.1a, the Raman peak position shows no change with varying MoS2 thickness, which contradicts numerous previous reports. The author should provide a detailed explanation for this discrepancy.

(2) Figure S2.1d does not appear to depict 3 nm-thick MoS2 layers; the surface looks uneven. This discrepancy should be addressed and explained.

(3) Regarding Figure 4, it is strongly recommended to include a comparison of photoluminescence (PL) spectra between reported monolayer MoS2 and bulk MoS2 with sulfurized MoO3 thin films to substantiate the authors' claim

(4) minor: check the arragement of figures and axis text, partcularly Fig. 1. Figures are not aligned. it make this figure unprofessional. 

Round 2

Reviewer 1 Report

Comments and Suggestions for Authors

I agree to publish this version of the manuscript.

Reviewer 2 Report

Comments and Suggestions for Authors

The author has successfully addressed all the issues raised in the previous review. Therefore, I recommend the current version of the manuscript for publication in Nanomaterials.